# Environmental Factors and the Risk of Developing Type 1 Diabetes—Old Disease and New Data

**DOI:** 10.3390/biology11040608

**Published:** 2022-04-16

**Authors:** Katarzyna Zorena, Małgorzata Michalska, Monika Kurpas, Marta Jaskulak, Anna Murawska, Saeid Rostami

**Affiliations:** 1Department of Immunobiology and Environment Microbiology, Faculty of Health Sciences with Institute of Maritime and Tropical Medicine, Medical University of Gdańsk, 80-210 Gdańsk, Poland; malgorzata.michalska@gumed.edu.pl (M.M.); monika.kurpas@gumed.edu.pl (M.K.); marta.jaskulak@gumed.edu.pl (M.J.); anna.murawska@gumed.edu.pl (A.M.); 2Department of Environmental Health Engineering, School of Health, Shiraz University of Medical Sciences, Shiraz P.O. Box 71348-14336, Iran; rostami_sa@sums.ac.ir

**Keywords:** incidence of type 1 diabetes, viruses, bacteria, yeast-like fungi, molds, climatic conditions, vitamin D deficiency

## Abstract

**Simple Summary:**

Despite many studies, the risk factors of type 1 diabetes (T1DM) in children and adolescents are still not fully understood and remain a big challenge. Therefore, an extensive online search for scientific research on factors related to diabetes has been performed for the identification of new factors of unexplained etiology. A better understanding of the role of viral, bacterial, and yeast-like fungi infections related to the risk of T1DM in children and adolescents and the identification of new risk factors, especially those spread by the droplet route, is of great importance for people and families with diabetes.

**Abstract:**

The incidence of type 1 diabetes (T1D) is increasing worldwide. The onset of T1D usually occurs in childhood and is caused by the selective destruction of insulin-producing pancreatic islet cells (β-cells) by autoreactive T cells, leading to insulin deficiency. Despite advanced research and enormous progress in medicine, the causes of T1D are still not fully understood. Therefore, an extensive online search for scientific research on environmental factors associated with diabetes and the identification of new factors of unexplained etiology has been carried out using the PubMed, Cochrane, and Embase databases. The search results were limited to the past 11 years of research and discovered 143 manuscripts published between 2011 and 2022. Additionally, 21 manuscripts from between 2000 and 2010 and 3 manuscripts from 1974 to 2000 were referenced for historical reference as the first studies showcasing a certain phenomenon or mechanism. More and more scientists are inclined to believe that environmental factors are responsible for the increased incidence of diabetes. Research results show that higher T1D incidence is associated with vitamin D deficiency, a colder climate, and pollution of the environment, as well as the influence of viral, bacterial, and yeast-like fungi infections. The key viral infections affecting the risk of developing T1DM are rubella virus, mumps virus, Coxsackie virus, cytomegalovirus, and enterovirus. Since 2020, i.e., from the beginning of the COVID-19 pandemic, more and more studies have been looking for a link between Severe Acute Respiratory Syndrome Coronavirus 2 (SARS-CoV-2) and diabetes development. A better understanding of the role of viral, bacterial, and yeast-like fungi infections related to the risk of T1DM in children and adolescents and the identification of new risk factors, especially those spread by the droplet route, is of great importance for people and families with diabetes.

## 1. Introduction

According to the World Health Organization (WHO), the number of people with diabetes worldwide increased from 108 million in 1980 to 422 million in 2014. The latest global estimate indicates that 40% of the 9 million patients diagnosed with T1D are under the age of 40. The prevalence is about 10 times higher in high-income countries compared to low-income countries. The dynamic increase in the number of cases found in children aged 0–4 years is also worrying [1]. The risk of developing the disease in the general population is 6%, while in family members of a person already suffering from it, the risk increases 15 times [2]. Despite many studies, the risk factors of type 1 diabetes in children and adolescents are not fully understood and remain a big challenge for researchers. Previous studies have shown that genetic and environmental factors have a significant impact on the risk of developing T1DM [3,4]. However, in the last few years, new factors have emerged that were not previously considered possible risk factors for the development of T1DM, such as particulate matter (PM) contamination, the microbiome, or SARS-CoV-2 virus infection [5,6,7]. Therefore, an extensive online search for scientific research on factors related to diabetes has been performed for the identification of new factors of unexplained etiology. Three databases were searched: PubMed, Cochrane, and Embase. The following ‘methodological filter’ was applied when surveying the literature. Firstly, a comprehensive search of peer-reviewed journals, excluding conference papers, was completed based on a wide range of key terms, including the following keywords: type 1 Diabetes Mellitus, T1DM, infection, viral infection, bacterial infection, fungal infection, microbiome, mold, environment, and environmental. Latin names of each of the pathogens found by the use of the previous keywords were also searched. Secondly, the reference section for each article found was also searched to find additional manuscripts. The search results were limited to the past 11 years of research and discovered 143 manuscripts published between 2011 and 2022. Additionally, 21 manuscripts from between 2000 and 2010 and 3 manuscripts from 1974 to 2000 were referenced for historical reference, as first studies showcasing a certain phenomenon or mechanism. Eligibility criteria chosen for the study were studies published in the English language between 2000 and 2022 focused on children, adolescents, or pregnant women (due to the T1DM manifesting itself in the first years of life). Exclusion criteria included: primary focus on type 2 diabetes with only a reference to type 1, short reports/conference articles without methodology or results section, studies published in a language other than English. Animal studies and cell-culture studies were only referenced to explain the mechanism behind a particular relation seen in human studies. 

### 1.1. Type 1 Diabetes Mellitus (T1DM)—Etiopathogenesis

During the initiation of the development of type 1 diabetes, under the influence of genetic and environmental factors, a cellular and humoral response is triggered against the β cells of the islets of Langerhans. Their slow destruction is the preclinical stage of diabetes, so-called “prediabetes”. The first clinical symptoms of the disease appear when 80–90% of pancreatic β-cells are destroyed. They appear suddenly and mainly include pollakiuria and polyuria as well as strongly increased thirst [8]. The symptoms of type 1 diabetes are accompanied by weight loss, fatigue, and weakness of the body. Ketone bodies may appear in the body as a result of a number of metabolic processes, leading to the development of diabetic ketoacidosis or keto coma, cerebral edema, and in extreme cases, death [8,9]. 

The clinical symptoms of type 1 diabetes mellitus are preceded by a long period of “prediabetes”, characterized by the presence of anti-islanding antibodies and impaired insulin and C-peptide secretion. During the “prediabetes” period, those antibodies detected in the blood serum are one of the markers for “predicting” the development of T1DM. These include antibodies against islet cells (ICA), insulin (IAA), and/or glutamic acid decarboxylase (GAD; an enzyme present in pancreatic β cells, participating in insulin secretion), against protein tyrosine phosphatase (anti-IA-2) and zinc transporting protein (ZnP8) [10]. Clinical symptoms of T1DM are preceded by disturbances in insulin and C-peptide secretion [10]. What is more, in patients with freshly diagnosed T1DM, autoreactive T lymphocytes directed against specific autoantigens, which are considered a cell-damaging factor, may appear in addition to serum antibodies [11]. In the course of viral infections or due to other environmental factors, autoreactive T cells can be activated and damage the β-cells of the pancreas [11,12]. β-cells are destroyed by apoptosis (programmed cell death) and necrosis. Both processes may involve cellular immunity, including lymphocyte T subpopulations (Th1 and Th2 helper and CD8 + cytotoxic cells), NK cells, monocytes/macrophages, cytokines, and other inflammatory factors [13,14]. According to the current knowledge, it is assumed that specifically activated Th1 lymphocytes, which recognize pancreatic autoantigens, induce migration of specific as well as non-specific mononuclear cells through cytokine secretion. These cells also produce pro-inflammatory cytokines that intensify the course of the reaction and, at the same time, affect β cells by stimulating the release of free radicals and nitric oxide [12]. The period in which the pancreatic islets are infiltrated by mononuclear cells, e.g., macrophages, T lymphocytes, and Natural Keller (NK) cells, is called perinsulitis [15].

Mononuclear cells and the release of inflammatory agents damage the β-cells of the pancreas by triggering the apoptosis process and by the infiltration of inflammatory cells called insulitis [16,17]. In the pathogenesis of T1DM, attention is also paid to the imbalance between the activity of the Th1 and Th2 lymphocyte subpopulations. Th1 lymphocytes present with features of autoreactive cells—they have the ability to secrete pro-inflammatory cytokines, e.g., IL-1, IL-6, IL-12, or TNFα, as well as activate macrophages, cytotoxic CD8 + lymphocytes and NK cells [18,19]. On the other hand, Th2 lymphocytes also act as regulatory cells—they can inhibit Th1 lymphocytes and induce a humoral response. In prediabetes patients, as well as in patients with newly diagnosed diabetes, there is an increased concentration of pro-inflammatory cytokines—the Th1 profile [20]. Studies on anti-inflammatory cytokines—Th2 profile of IL-4, IL-10, IL-13 have shown that the ability to secrete these cytokines may correlate with a reduced risk of developing diabetes [10,16,20]. On the other hand, under the influence of pro-inflammatory factors (IL-1, TNF-α, and IFN-γ), the production of NO intensifies, which in turn may lead to apoptosis of pancreatic β-cells [21,22]. 

### 1.2. Genetic Predisposition as an Etiological Factor of T1DM 

T1DM is a multifactorial disease. Susceptibility to T1DM is hereditary, and the mode of inheritance is complex and still not fully understood [9]. The risk of developing T1DM under the age of 20 is 1:300 in families without genetic burden, 1:50 in the case of a child whose mother has T1DM, and 1:15 in the case of T1DM in the child’s father. A family history of diabetes is an important indicator of the risk of the disease in children because it shows the coexistence of pro-diabetogenic factors, both genetic and environmental [23,24]. The main genes predisposing to the development of T1DM are located within the genes of the HLA system on chromosome 6, although other genes may also be involved [25,26]. The genetic factor plays a significant role in the development of T1DM, which has been proven by studies on twins. In identical twins, the risk of diabetes is 50% compared to fraternal twins, where the risk ranges from 5 to 13% [27,28]. A study by *Siewko* et al. showed that autoimmune markers were detected in the serum in more than 30% of first-degree relatives of diabetic patients. It was also revealed that these people, despite normal glucose tolerance, significantly decreased beta-cell secretory reserve and decreased insulin sensitivity [29].

The genetic predisposition to develop T1DM is associated with the HLA gene complex located in chromosome 6p21 (IDDM1), which has been known since the 1970s. [30]. The associated HLA DR3 and DR4 haplotypes cause severe susceptibility [31,32]. In 90% of the Caucasian population, diabetes is strongly associated with DR3 antigens (DRB1 * 03: 01-DQB1 * 02: 01) and DR4 (DRB1 * 04: 01-DQB1 * 03: 02). In contrast, in Japan and most other East Asian populations, TDM1 diabetes is associated with DR4 haplotypes (DRB1 * 04: haplotypes 05-DQB1 * 04: 01) and DR9 (DRB1 * 09: 01-DQB1 * 03: 03) [33]. In Brazil, the most common haplotype is HLA-DRB1 * 03: 01 ~ DQA1 * 05: 01 g ~ DQB1 * 02: 01 [32]. 

Analysis of the human genome by a mapping technique also identified other loci attributed to the presence of T1DM. One such site is the region associated with the insulin gene, which is located on the short arm of chromosome 11 (11p15.5) and is responsible for 10% inheritance of T1DM [34]. So far, the risk of T1DM has been associated with nearly 30 different gene loci. However, few of the loci studied have been linked to specific genes [34]. Research on the connection of genes with a genetic predisposition is still ongoing. Despite many years of research, the actual role of most of these genes in the pathogenesis of T1DM remains unclear [35,36,37]. On the other hand, more and more studies indicate the role of environmental factors in the development and/or progression of T1DM [6,38,39,40,41,42].

### 1.3. The Accelerator Hypothesis, the β-Cell “Overload” and the Risk of Developing Diabetes

Genetic inheritance is linked to various hypotheses with environmental factors and is associated with multiple mechanisms [43,44,45,46,47,48]. One of these hypotheses is the “Accelerator Hypothesis”, which points to the role of adipose tissue metabolism in the process of autoimmunity. Proposed by Kibirige et al. and based on epidemic logical data, the hypothesis suggests that excess body fat and insulin resistance may be accelerators, i.e., they may accelerate the destruction of β-cells (by the primary or secondary effect on their apoptosis) and, consequently, initiate the autoimmune process in genetically predisposed people [49]. Moreover, it was shown that low physical activity had been shown to initiate insulin resistance leading to excessive β-cell function [50]. The increasingly ‘obesogenic’ environment which promotes insulin resistance could account for the rising incidence of type 1 diabetes [51].

The second known hypothesis is the so-called “overload of the β cell” theory formulated by Dahquist G., which is a complement to the accelerator hypothesis [52]. It assumes the influence of several environmental factors on the increase of β-cells susceptibility to necrosis and/or apoptosis, including factors such as overfeeding the fetus and children. Interestingly, recent research has shown that excessive weight gain in the first years of life causes growth and the accumulation of both fat and muscle cells [53]. On the other hand, recent research does not corroborate an increase in type 1 incidence in the pediatric population being associated with younger age of diagnosis and higher BMI-SDS [54]. The data does not support the ‘accelerator hypothesis’. There was no sign of excessive weight gain before the manifestation of type 1 diabetes. The authors suggest that discrepant results from other studies could be due to non-age-adjusted controls [55].

## 2. Environmental Factors

### Climatic Conditions, Lack of Vitamin D Deficiency, Feeding with Cow’s Milk, and the Risk of Developing Type 1 Diabetes

Over the past decade, there has been a significant increase in scientific research into the effects of vitamin D on human health. In addition to its well-established role in regulating calcium metabolism, vitamin D deficiency is associated with the risk of many chronic diseases, including T1DM [56,57,58,59,60]. It is worth noting that 1.25 (OH) (2) D regulates the expression of over 200 different genes, including those related to apoptosis and immune modulation [59,61,62]. The significance of vitamin D deficiency and lack of sun exposure in the development of diabetes is confirmed by studies of children living in Norway and Denmark [63,64]. In Norway, the relation between 25 (OH) D levels in pregnant women and the risk of developing T1DM in newborns was investigated. The study was conducted on a population of 220 women. It showed that in the first and second trimesters, the results of 25 (OH) D levels did not differ significantly. However, in the third trimester, the level of 25 (OH) D was lower (0.27 μmol/L [(95% CI, 0.57, 0.03)] than in the control group (5.01 nmol/L [(95% CI, 8.03, 0.73)], which may be associated with a higher risk of developing diabetes in the offspring. Danish authors conducted a study on the exposure of pregnant women to sunlight. The study included 331,623 people born in the years 1983–1988, of which 886 children (0.26%) developed TDM1 at the age of 15. In contrast, 503 children were diagnosed with TDM1 at the age of 10–15. The study showed that higher sun exposure in the third trimester of pregnancy was associated with a lower risk (HR) of T1D at 5 to 9 years of age in boys: HR (95% CI): 0.60 (0.43–0.84), *p* = 0.003 [64]. Studies have also reported the effects of vitamin D supplementation during pregnancy on postpartum glucose metabolism. In their research, Valizadeh et al. stated that prenatal vitamin D supplementation in patients with gestational diabetes and vitamin D deficiency safely increases the level of 25OHD in both mother and child. The authors showed that the increase in vitamin D level was maintained for several weeks postpartum but did not significantly affect fasting glucose, insulin levels, and insulin resistance [65]. Other authors have shown that vitamin D deficiency in pregnant women occurs in many regions of the world, whether they come from Mediterranean countries or from countries with higher latitudes in Central and Western Europe [66]. Moreover, skin color is also one of the parameters determining vitamin D deficiency during pregnancy. Pregnant women with darker skin pigmentation have a greater risk of developing vitamin D hypovitaminosis than white women [66]. Additionally, Mohr et al. found that the low intensity of UVB radiation at higher latitudes in both hemispheres (R2 for latitude = 0.25, *p* < 0.0001) and the number of hours of sunshine per day were associated with a much higher incidence of T1D in childhood [67]. Other researchers, including Cherrie et al., indicated that people living closer to the coast had higher vitamin D levels, especially in the fall, than those living inland [68]. Also, research conducted by the Abel and Fava team showed a negative correlation between the distance from the sea and the incidence of TDM1. Researchers suggest that environmental factors such as sunlight, air temperature, and latitude related to climatic conditions may influence the risk of TDM1 in children [69].

The case-control study included 101 children (41 boys and 60 girls) under 16 years of age, from which the investigators saw a relationship between maternal age at birth, cow’s milk feeding, lack of vitamin D supplementation, cesarean section, and the development of T1DM in children [70]. Selected studies on the impact of climate, vitamin D deficiency, and cow’s milk feeding on the risk of developing type 1 diabetes are presented in Table 1.

Egyptian researchers have conducted studies on the potential influence of environmental factors on the development of T1DM in children. The study included 204 children aged 6 to 16 years with diabetes mellitus type 1. The control group consisted of 204 healthy children of the same age. A gender-adjusted multivariate logistic regression model revealed that the risk of developing T1DM was significantly higher among children living in the countryside (aOR = 2.03, 95% CI: 1.30–4.25), those with parents who had T1DM (aOR = 9.03, 95% CI: 1.02–83.32), those delivered by cesarean section (aOR = 2.13, 95% CI: 1.09–5.03), and those who were fed cow’s milk during the first years of life (aOR = 19.49, 95% CI: 8.73–45.53). On the other hand, a protective effect has been observed between breastfeeding for at least six months, vitamin D supplementation in the first year of life, high physical activity, and the development of T1DM in children [71]. In light of the existing knowledge, it has been proven that the autoimmune process is associated with the need for an environmental factor to initiate the autoimmune process, but it is not a rule. The immune reaction in the pancreas may last from several weeks to several years, and the manifestation of diabetes itself is associated with severe stress, the period of sexual maturation, and most often with bacterial and viral infections [72,73,74,75]. The effect of bacteria, viruses, and fungi on the risk of developing diabetes and chronic complications has been tested in many centers in Poland and around the world [74,76,77,78,79,80,81,82,83]. However, proving a cause-and-effect relationship between bacterial and viral infections and the occurrence of T1DM is extremely difficult [75,84]. The main problem is the often long period between exposure to a given antigen of a virus, bacteria, or yeast-like fungal cells and the onset of clinical symptoms of T1DM [85,86]. Seasonality of the disease has also been demonstrated in children and adolescents suffering from type 1 diabetes. In Europe, the number of cases increases significantly in the fall and winter months, which may be related to viral infections, which are quite common at that time [87,88].

## 3. Viral Infections and the Risk of Developing T1DM

So far, the most important environmental factors have been viral infections of rubella, Coxsackie, mumps, and enterovirus. Many years of research also indicate a significant influence of adenoviruses or the Epstein–Barr virus [89,90,91,92,93]. In the last two years, more and more studies have indicated the important role of the SARS-CoV-2 virus in the development and progression of type 1 diabetes [75,79,85,94,95,96]. The possible influence of environmental factors on the risk of developing type 1 diabetes mellitus is shown in Figure 1.

There are two main known mechanisms behind virus-induced damage to pancreatic β-cells. On the one hand, it may be a direct cytolysis of virally infected cells without the involvement of the immune system, but on the other hand, it may be a viral induction of autoimmune phenomena [12]. Some similarities were revealed between the amino acid sequence of the β-cell-specific glutamic acid decarboxylase (GAD) molecule and the abundance of proteins derived from cytomegalovirus, Epstein–Barr, Coxsackie, and adenovirus. Based on in vitro studies, it has been shown that viral infection directly affects the secretion of pro-inflammatory mediators (TNF-α, IFN, NO), leading to the destruction of pancreatic β-cells [44,45,46,47,48,49,50,51,52,53,54,55,56,57,58,59,60,61,62,63,64,65,66,67,68,69,70,71,72,73,74,75,76,77,78,79,80,81,82,83,84,85,86,87,88,89,90,91,92,93,94,95,96,97]. In the case of viral induction of autoimmune phenomena leading to the development of diabetes mellitus, the most common theory is the molecular mimicry model. The similarity of particle fragments derived from exogenous proteins (viral proteins) with β-cell antigens may contribute to the penetration of autoreactive T cells into the bloodstream and the initiation of the autoimmunity process, leading to the destruction of the pancreatic islets. This mechanism is still not well understood, GAD is certainly one of the autoantigens, but antibodies to insulin (IAA), tyrosine phosphatase (IA2), and GLUT-2 protein may also appear [98]. For virions in the host organism to be destroyed, the viral peptides must be presented by macrophages. If the presented antigens show molecular similarity to the organism’s antigens, autoreactive B and T lymphocytes are produced. The consequence of this may be a cross-reaction of these cells with virions and autoantigens on the surface of pancreatic cells [98].

Viruses can cause type 1 diabetes by infecting pancreatic beta cells, causing direct cytotoxicity, or inducing an autoimmune response against beta cells [99,100]. However, establishing consistency in findings across studies has proven difficult. Obstacles to convincingly linking RNA viruses to islet autoimmunity may be attributed to rapid viral mutation rates, the cyclical periodicity of viruses, and the selection of variants with altered pathogenicity and ability to spread in populations [90,91,92,93,94,95,96,97,98,99,100,101]. In studies conducted by Krogvold et al., enteroviruses were detected in the pancreas of patients with newly diagnosed type 1 diabetes [86]. Also, the research by Hodik et al. confirmed the increased level of CAR receptor (Coxsackie and adenovirus receptor) expression on pancreatic cells in patients diagnosed with type 1 diabetes [101]. In the research work of Honkanen et al., the material for the isolation of enterovirus strains was feces collected from children aged 3 months to 2–3 years. The studies confirmed the observations made by other scientists that enteroviral infections may play a role in initiating the process of autoimmune destruction of beta cells. Researchers also found that viral infections occurred even a year before insulin production disorders were diagnosed [90].

Vehik et al. showed that prolonged enterovirus B rather than independent, short-duration enterovirus B infections might be involved in the development of islet autoimmunity, but not T1D, in some young children [91]. Moreover, metagenomic sequencing was performed on fecal specimens from 383 islet autoimmunity and 112 T1D case-control matched pair children from 6 TEDDY study sites distributed in the US, Germany, Sweden, and Finland. Samples were collected approximately monthly, from age 3 months until case event-time, totaling 8654 stools for the islet autoimmunity and 3380 stools for the T1D nested-matched case-control studies. The researchers also found that young children were less infected with human mast adenovirus C and that the CXADR rs6517774 receptor was independently correlated with islet autoimmunity [91].

Recently, a clinical case of a 31-year-old pregnant patient suspected of having type 1 diabetes was described (Fulminant type 1 diabetes mellitus). Tests for the presence of antibodies in the patient’s serum confirmed infection with Coxsackievirus B1 [92]. American scientists, on the other hand, conducted in vitro studies confirming whether human β stem cells (SC-β) could serve as a model for studying the effect of viral infection on TDM1 [102]. The human β stem cells (SC-β) used for the study were derived from the embryonic stem cell line HUES8 and were infected with the Coxsackie virus strain JVB CVB4. Studies have shown that SC-β cells can serve as a model for T1DM development. The same authors proved that CVB infection strongly induces innate immune responses, transcriptional decreases in insulin, and impaired insulin secretion [102]. In vitro studies, on the other hand, suggest that duct cells may be the site where enterovirus survives, allowing it from time to time to spread to pancreatic β-cells. However, the authors of the studies make a reservation that further studies are needed to assess the presence of viruses in the pancreas in order to further test the presented hypothesis [103]. In other studies, the authors showed a strong correlation between the cytomegalovirus (CMV) genome detectable in lymphocytes and autoantibodies against pancreatic islets in patients with newly diagnosed type 1 diabetes [104]. Yoned et al. carried out a histopathological examination of the pancreas within a study that showed damage to β-cells as a result of CMV infection [105]. Another study by Saber and Mohammed also indicated a significant role of cytomegalovirus and Epstein–Barr virus in the pathogenesis of type 1 diabetes [47]. The researchers studied 56 patients aged 3 to 22. Out of those, 53 (94.60%) patients with T1DM were positive for IgG antibodies against cytomegalovirus, and 24 (42.90%) for IgG antibodies against Epstein–Barr virus compared to the control group [47]. On the other hand, Ekman et al., prospectively testing 1402 children for IgG-specific antibodies against CMV, found that a history of CMV infection in early childhood may slow down autoimmunity of pancreatic islets and, thus, may protect these children against the onset of TDM1 during childhood [106]. Interestingly, AL-Hakami et al. did not find any correlation between the presence of CMV IgG antibodies in children diagnosed with type 1 diabetes [107]. Similarly, Aarnisalo et al. found no relation between perinatal CMV infection in early infancy and the appearance of autoimmune beta cells or the progression of type 1 diabetes in children with the HLA-DQB1 genotype [104]. 

Another virus species suspected of causing type 1 diabetes is Parvovirus B19 (B19V), belonging to the Parvoviridae family, genus Erythroparvovirus. The clinical case described by Nishiumi et al. concerning a patient with an increased concentration of anti-parvovirus B19 IgM in the blood serum confirmed the relationship between Parvovirus B19 infection and fulminant TDM1 [108]. Moreover, Eklioğlu et al. described the case of a 9-year-old boy who was diagnosed with type 1 diabetes 3 years earlier and who was hospitalized due to ketoacidosis (DKA) and acute hepatitis caused by B19 parvovirus [109]. However, other research involving 22 children aged 1 to 17 years with newly diagnosed type 1 diabetes did not show any correlation between T1DM onset and Parvovirus B19 infection [110]. Rotavirus (RV) remains the major cause of infantile gastroenteritis worldwide. Rotavirus infections were initially identified as possible triggers of T1D, given similarities between viral peptide sequences and T1D autoantigen peptide sequences [111,112,113].

The research by Honeyman et al. confirmed the relationship between infection by rotavirus A (RV), which causes childhood gastroenteritis, and the incidence of T1D [111]. The most recently published studies show that rotavirus vaccination may be associated with a decline in T1DM1 incidence. For example, the incidence of T1DM in Australian children under 5 years of age has decreased following the introduction of the rotavirus vaccine [114]. However, previous Finnish studies have shown no relationship between the oral rotavirus vaccine and the risk of T1DM in children over 4–6 years of follow-up [115]. Additionally, studies by Glanz et al. showed no correlation between the oral rotavirus vaccine and the risk of T1DM. Researchers conducted a study in the US of a cohort of 386,937 children vaccinated against RV—360,169 (93.1%) with the full vaccine; 15,765 (4.1%) with the incomplete vaccine; and 11,003 (2.8%) children not vaccinated against RV at all [116]. Other studies from this large cohort study did not provide evidence that rotavirus vaccination prevents CD or T1D, nor is it associated with increased risk, delivering further evidence of rotavirus vaccine safety [117].

Other viral factors that may play a role in the pathogenesis of T1DM also include rubella, mumps, and measles viruses [118,119,120]. Infection with rubella virus during pregnancy is associated with an increased risk of diabetes in the offspring. Type 1 diabetes mellitus has been shown to develop in 12–20% of patients with congenital rubella infection, and 40% of patients have abnormalities in the oral glucose tolerance test [119]. Genetic studies also showed that patients with congenital rubella syndrome who developed T1DM had a significant increase in HLA-DR3 antigens and a decrease in the HLA-DR2 haplotype. In addition, autoantibodies against insulin and cytoplasm of pancreatic islet cells and abnormalities in T cells were detected in T1DM patients.

Ramondetti et al. showed a relation between the number of measles, mumps, and rubella cases and the number of children with T1DM from 1996 to 2001. Italian researchers conducted studies on children aged 0–14 years with newly diagnosed type 1 diabetes. They observed a significant relation between the incidence of type 1 diabetes and the incidence of rubella (*p* = 0.014). However, they did not find a statistically significant relation between the incidence of measles and the incidence of type 1 diabetes in children (*p* = 0.269) [118]. In Turkey, the clinical case of a 13-year-old boy with TDM1 who was diagnosed with congenital rubella syndrome was described [119].

The virus that can destroy the β-cells of the Langerhans islets and, consequently, lose the ability to secrete insulin is the mumps virus. As a result of infection with mumps virus, the expression of HLA class I is disturbed, and the accompanying malfunction of T lymphocytes, which triggers the processes of increased expression of the HLA I molecule, and may affect the autoimmune process in people in the prediabetic period. Researchers in Italy observed a significant relation between the incidence of type 1 diabetes in children aged 0–14 years and the incidence of mumps (*p* = 0.034). The correlation between mumps virus infection and diabetes was also investigated using an in vitro model. Islet cell clusters (ICC) prepared from human fetal pancreas were infected with both mumps virus and Coxsackie B3 virus. The studies showed that viruses infected the insulin-secreting cells and other cells in the pancreas and that secretion of immunoreactive insulin into the culture medium of mumps virus-infected cells ceased on day 7. Mumps virus-infected ICCs were detected for 14 days, and mumps virus antigen was detected in ICCs throughout the 22-day observation period [121]. 

There are also indications that human endogenous retroviruses (HERV) are associated with type 1 diabetes, multiple sclerosis, and amyotrophic lateral sclerosis. Endogenous human retroviruses can induce the production of pro-inflammatory cytokines such as IL-1, IL-6, or TNF-α by cells such as TDM1 triggering monocytes [122]. Levet et al. detected HERV-W-Env protein in 70% of patients with type 1 diabetes, and the corresponding RNA was detected in 57% of peripheral blood mononuclear cells. Studies of human islets of Langerhans showed inhibition of insulin secretion by HERV-W-Env. Scientists found that this endogenous protein is expressed in acinar cells of the pancreas in 75% of TDM1 patients. In immunohistological studies, the same study also showed a significant correlation between the expression of HERV-W-Env and macrophage infiltration from the exocrine part of the pancreas. These results have been confirmed in in vivo tests. Hyperglycaemia, decreased insulin levels, and infiltration of immune cells in the pancreas have been detected in studies with transgenic mice. The obtained results suggest the participation of HERV-W-Env in the pathogenesis of TDM1 [123]. 

It is also hypothesized that *Mycobacterium avium* subsp. *paratuberculosis* (MAP) may be a potential risk factor for developing type 1 diabetes in humans [124,125]. French researchers investigated the humoral response to the HERV capsular antigens (HERV-KEnv and HERV-WEnv) and four MAP-derived peptides with human homologs in pediatric populations. In work by Negowska et al., the authors presented a study of 142 children at risk of developing TDM1, including 14 children from Sardinia, children from mainland Italy (*n* = 54), adolescents from Poland (*n* = 74), and adolescents with obesity not associated with autoimmunity (*n* = 26). The research results confirmed the hypothesis of MAP infection affecting the expression of the HERV-W antigen and increasing the production of autoantibodies in TDM1 [125]. 

The latest reports have also shown that in the current COVID-19 pandemic, more and more scientists confirmed the fact that the new SARS-CoV-2 virus may affect the development of T1DM. The role of COVID-19 in the development of T1DM is not yet fully understood, but research results indicate that pancreatic cell damage may occur, especially in patients with severe SARS-CoV-2 infection [79,94,96,126,127]. On the other hand, it is already confirmed that diabetes affects the much more severe symptoms of COVID-19 [128,129]. Although diabetes is an important risk factor in the case of COVID-19 infection, some studies indicate that age is also essential in patients with diabetes and COVID-19. The studies performed by Demeterco-Berggren et al. showed that patients over 40 years with diabetes are more exposed to severe COVID-19 [129]. 

The fact is that in both the SARS-CoV epidemic and the SARS-CoV-2 pandemic, the presence of the virus was demonstrated in the pancreas of deceased patients [130,131]. Research conducted by a Chinese team of researchers during the SARS-CoV coronavirus epidemic showed a significant link between pancreatic cells and the presence of the virus’s genetic material. Analyzes were performed on post-mortem material using RT-PCR and focused on the detection of the M protein genes [131]. Similarly, research was also carried out during the current pandemic by scientists from Germany [130]. Based on post-mortem studies of 11 patients who died from COVID-19, the presence of SARS-CoV2 genetic material in the pancreas of COVID-19 patients was also confirmed [130].

Another important aspect related to COVID-19 in the context of diabetes mellitus is the fact that SARS-CoV 2 infection may exacerbate symptoms associated with diabetes complications (vasculopathy, coagulopathy). This is due to an imbalance in the activation pathways of vascular converting enzymes (ACE2), which influences the development of the body’s inflammatory response. These metabolic disorders cause beta-cell dysfunction and glycemic status [95]. Studies also indicate a more severe course of COVID-19 in patients with diabetes mellitus and lower effectiveness of insulin treatment [72]. The results of these studies may suggest a link between COVID-19 and damage to pancreatic beta cells, which in the longer term may affect the development of type 1 diabetes in patients under COVID-19. Observations so far provide support for the hypothesis of a potential diabetogenic effect of COVID-19 beyond the well-recognized stress response associated with severe illness. However, whether the alterations of glucose metabolism that occur with a sudden onset in severe COVID-19 persist or remit when the infection resolves is unclear. That is why an international group of leading diabetes researchers participating in the CoviDIAB Project have established a global registry of patients with COVID-19–related diabetes (covidiab.e-dendrite.com). The goal of the registry will be to establish the extent and phenotype of new-onset diabetes that is defined by hyperglycemia, confirmed COVID-19, a negative history of diabetes, and a history of a normal glycated hemoglobin level. The registry, which will be expanded to include patients with preexisting diabetes who present with severe acute metabolic disturbance, may also be used to investigate the epidemiologic features [79]. The likely influence of viruses on the risk of developing T1DM is presented in Table 2.

## 4. The Effect of Bacteria and Yeast-Like Fungi on the Risk of Developing T1DM Diabetes

One of the elements that have a significant impact on the quality of external and internal air is the presence of microorganisms that are part of the bioaerosol. The biological pollutants of the air are mainly pollen, fungal spores, bacteria, and viruses [134,135,136]. Indoor air quality, on the other hand, depends both on factors related to the external environment and factors occurring in the internal environment. In indoor spaces, one of the main sources of bioaerosol is humans (sweat droplets, saliva, sneezing, coughing). When sneezing, a person expels up to 40,000 droplets at a speed of 100 m/s, and coughing can produce about 3000 droplets that are transported over long distances [137,138]. The extent to which biological particles suspended in the air are inhaled depends significantly on their ability to penetrate the respiratory system and their deposition within it. Particles larger than 10 μm in diameter can form deposits in the upper respiratory tract, nose, and throat, causing hay fever. Particles smaller than 5 μm in diameter penetrate alveoli, causing allergic alveolitis and asthma. Bound and non-bound fungal allergens, ultra-small particles measuring less than <0.1 μm, can deeply penetrate the human respiratory system. Their size and shape also determine their removal from the airways [139]. Biological particles suspended in the air may not only be a direct cause of allergies, e.g., allergic alveolitis and asthma, but also the etiological factors of many other diseases, such as bronchitis and pneumonia, rhinitis, diabetes, and neoplastic diseases [74,135,140]. Infection with mold spores can occur by inhalation, damaged skin, or mucosal damage. The most common fungal infections observed in diabetic patients are aspergillosis, mucormycosis, and hyalohyphomycosis. There were also described cases where the clinical course, radiological changes, and mycological tests met the criteria of chronic necrotic pulmonary aspergillosis (CNPA) caused by *Aspergillus niger* and fumigatus fungi [132]. Allergic bronchopulmonary aspergillosis (ABPA) is also most often caused by infection with *Aspergillus fumigatus*. In addition, it can also occur after infection with other fungi, including *Culvularia* sp., *Candida* sp., *Aspergillus niger*, *A. flavus*, *A. nidulans*, *A. orizae*, *A. glaucis* [133,141,142]. Dayal et al. detected mucormycosis in children aged 3 to 12 years with T1DM and diabetic ketoacidosis (DKA) [143]. The diagnosis of mucormycosis was based on microbiological and histopathological examinations. The children were brought to the hospital with symptoms such as facial swelling, pain, fever, nasal congestion, and nasal discharge suggestive of severe sinusitis. A similar case was reported by Pennell et al. [144]. The authors diagnosed a case of pulmonary mucormycosis in a 13-year-old girl who was diagnosed with type 1 diabetes and ketoacidosis (DKA). The patient reported to the hospital with complaints of fever, nasal congestion, and increased cough in the preceding four weeks [145]. The threat is posed not only by the presence of mold spores, pathogenic microorganisms, and their toxins in the air but also by the excessive number of saprophytic microorganisms, especially if their composition is not very diversified and organisms of one species dominate. In Poland, in the Lubelskie Voivodeship in a study conducted from January 2015 to December 2016 found a relation between the number of new cases of T1DM and the number of psychrophilic bacteria (β = 2.86; *p* < 0.05), mesophilic bacteria (β = 2.824; *p* < 0.05), and the number of mold fungi (β = 2.923; *p* < 0.001). Additionally, studies conducted in the same period in Pomorskie Voivodeship showed a relationship between the number of psychrophilic bacteria (β = 2.96; *p* < 0.001) and mesophilic bacteria (β = 2.898; *p* < 0.001). However, no correlation was found between the number of new cases of T1DM and the number of molds (β = 2.756; *p* = NS) [74]. The authors suggest that there is a higher concentration of microbial particles in the Lublin Voivodeship. That is why they recommend changes in climate for children with T1DM (trips to the sea, mountains, etc.) as often as possible. The possible effects of bacteria and yeast-like fungi on the risk of developing T1DM are shown in Table 3.

The correlation between diabetes and bacteria is not only observed in the context of infection. Many studies, conducted both on the human subjects and mice models, indicate a connection between the composition of the intestinal microbiome and diabetes, and they focused on both the composition and diversity of microorganisms groups [143,144,145]. Research carried out by scientists indicates several potential mechanisms that may explain the impact of changes in the diversity of bacteria in the digestive system and the development of diabetes. However, the exact impact is very complex and has not yet been identified. The most important factors influencing the changes in the microbiome are antibiotic intake, drugs therapy, diet, pH of drinking water, and delivery mode [146,147]. The composition of the microbiome varies throughout life. It is indicated that in the case of newborns and infants, the delivery mode and method of feeding (breastfeeding or infant formula milk) have a great impact on intestinal microbiome diversity. Children born by cesarean section have a microbiome composition similar to the mother’s skin [148]. Research on the development of the human microbiome in infants and the correlation with the development of type 1 diabetes was conducted by Kostic et al. [149]. They showed that in the case of progression of the T1D, the number of microorganisms associated with inflammation increases. They also observed differences in the case of patients with progress and nonprogress of T1D [149]. In the next stages of life, there are more factors influencing the presence of bacteria in the human digestive system. Inadequate diet or drugs can significantly affect microbiome disturbance called dysbiosis, which affects higher insulin resistance and inflammation [150]. Differences in the composition and diversity of microorganisms in the digestive system were observed both in the cases of bacteria and fungi in studies conducted with patients with T1DM. Research conducted by Kowalewska et al. showed that patients with T1DM show higher species diversity of yeast-like fungi in stool samples. Moreover, fungal species in patients with T1DM turn out to be more resistant to antifungal treatment [83]. Research on the intestinal microbiome in the context of T1DM focuses on three main issues: reduction of the diversity of microbiota, disturbances in the Firmicutes/Bacteroides ratio, and the reduced number of probiotic bacteria such as lactobacillus [151]. In the studies carried out by Patterson et al., it was also shown that dysbiosis might manifest itself in a reduced differentiation of the microbiome, which in turn increases the predisposition to T1DM development [152]. Moreover, numerous studies show that the composition of the gut microbiome can influence the metabolism of carbohydrates, fatty acids, or amino acids which indirectly may impact the formation of metabolic disorders and diabetes [153,154,155,156]. The development of type 1 diabetes is largely correlated with the diet we eat. A diet high in sugars and fats can accelerate or even induce the changes that cause this disease [157]. Studies conducted by Calcinaro et al. show that supplementing the diet of non-obese mice with oral probiotics may inhibit the development of autoimmune diabetes [158]. Increasing inflammation is also an important factor in developing diabetes. Studies conducted by Pahwa et al. show a significant correlation between the composition of the gut microbiome and inflammation in the presence of diabetes and diabetic complications [159]. Studies on rat models also show a relation between diabetes and the composition of the gut microbiome. In studies published by Roesh et al. it was observed that in the case of bio-breeding diabetes-prone (BB-DP) rats, higher numbers of bacteria of the *Bacteroides*, *Ruminococcus,* and *Eubacterium* families were observed in stool samples [160]. A higher number of bacteria that may have a probiotic (*Lactobacillus* spp. *Bifidobacterium* spp.) effect was observed in bio-breeding diabetes-resistant (BB-DR) rats [161]. An important mechanism with a potential influence on the development of type 1 diabetes is the molecular mimicry of proteins produced by intestinal bacteria. Some of the proteins produced by the intestinal microbiome have a molecular structure similar to self-anti-gene in the pancreas. An example of such protein is the Mpt protein produced by the *Leptotrichia goodfellowii* bacteria of the Fusobacteria family [162].

### Limitations and Recommendations

The present systematic review has some limitations that we would like to address. Firstly, our review did not include patients with type 2 diabetes and other types of diabetes. Secondly, the manuscript would be more powerful if a meta-analysis was performed.

Regarding the recommendations, we believe that demonstrating the relationship between environmental pollution, infectious diseases (viral, bacterial, fungal), and the presence of type 1 diabetes makes it necessary to consider broader activities related to the prevention of T1DM. For viral infections such as measles, rubella, and mumps, vaccines are the most effective way to fight and control them [163,164]. In the context of reports about new outbreaks of these viral diseases, both in children and adults, the situation is becoming disturbing. It is estimated that this increasing number of cases is influenced by anti-vaccination movements and disinformation on the internet [165]. Therefore, further research and increased preventive measures are necessary, especially with vaccinations and educating the public in the context of all possible consequences of developing rubella, measles, and mumps. It is also necessary to take a holistic approach to the links between environmental pollution, viral diseases, and diabetes, particularly due to the fact that the content of PM2.5 and PM10 in the air is also associated with the occurrence of mumps, as indicated by studies carried out in China [166]. The second important recommendation is to pay attention to diet and the influence of the gut microbiome. Introducing more products containing probiotic bacteria in order to diversify the gut microbiome in the diet appears to be of particular importance [151,162,167] Therefore, due to the complex mechanisms influencing the development of type 1 diabetes, a holistic approach is important, with particular emphasis on environmental pollution, microbiology, immunobiology, and dietetics.

## 5. Conclusions

Despite advanced research and tremendous advances in medicine in recent years, the causes of type 1 diabetes are still unknown. In addition to genetic predisposition and immunological factors, lifestyle and environmental factors are considered to have an effect on the development of type 1 diabetes. Scientific sources also report that vitamin D deficiency, a cooler climate, early administration of cow’s milk, and gluten are also associated with more frequent type 1 diabetes. Since 2020, i.e., from the beginning of the COVID-19 pandemic, more and more studies have been looking for a link between SARS-CoV-2 and diabetes development. Observations so far provide support for the hypothesis of a potential diabetogenic effect of COVID-19 beyond the well-recognized stress response associated with severe illness. However, whether the alterations of glucose metabolism that occur with a sudden onset in severe COVID-19 persist or remit when the infection resolves remains unclear. A better understanding of the role of viral, bacterial, and yeast-like fungi infections related to the risk of T1DM in children and adolescents and the identification of new risk factors, especially those spread by the droplet route, is of great importance for people and families with diabetes.

## Figures and Tables

**Figure 1 biology-11-00608-f001:**
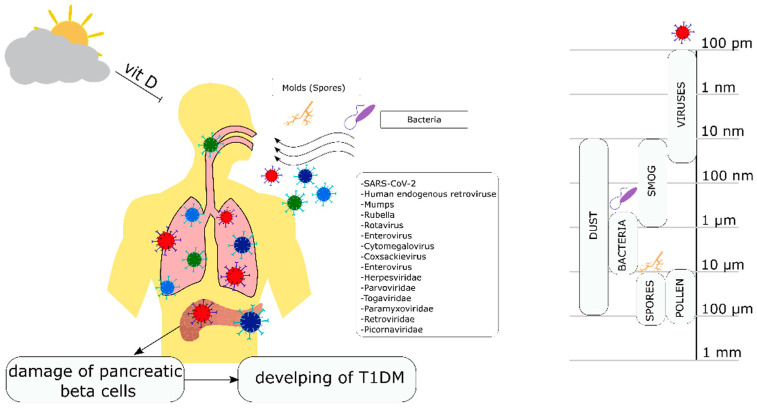
The possible influence of environmental factors on the risk of developing type 1 diabetes mellitus.

**Table 1 biology-11-00608-t001:** The climatic conditions, vitamin D deficiency, cow’s milk feeding, and the risk of developing type 1 diabetes.

No.	Authors	Climatic Conditions, Lack of Vitamin D Deficiency, Cow’s Milk Feeding	Consequences
56	Waernbaum and Dahlquist, 2016	Climate/air temperature and the number of sunshine hours	Low mean temperature rather than few sunshine hours are associated with an increased incidence of type 1 diabetes in children.
63	Sørensen et al., 2016	Vitamin D deficiency in pregnancy	Low level of 25 (OH) D during the third trimester of pregnancy may be associated with a higher risk of developing diabetes in the offspring.
64	Jacobsen et al., 2016	Sun exposure/Vitamin D deficiency in pregnancy	Higher sun exposure in the third trimester of pregnancy is associated with a lower risk (HR) of T1D at 5 to 9 years of age.
67	Mohr et al., 2008	Intensity of UVB radiation and the number of hours of sunshine per day	The low intensity of UVB radiation at higher latitudes in both hemispheres (R2 for latitude = 0.25, *p* < 0.0001) and the number of hours of sunshine per day were associated with a much higher incidence of T1D in childhood.
69	Abela and Fava, 2019	Climatic conditions	A significant negative association was found between T1D incidence and shortest distance from sea, mean hours of sunshine, and mean temperature, and a positive association was found between T1D incidence and latitude.
70	Ahadi et al., 2011	Association between environmental factors and risk of type 1 diabetes	Maternal age > 35 years at delivery, duration of > 6 months of cow milk feeding, lack of vitamin D supplementation, and cesarean delivery were significantly associated with diabetes type 1.
88	Kostopoulou, et al., 2021	Seasonal variation of T1DM onset	Seasonality of the T1DM diagnosis is shown, with a predominance in the colder months of the year.

**Table 2 biology-11-00608-t002:** The role of viruses in the development of type 1 diabetes.

No.	Authors	Viruses	Consequences
Viruses			
[23]	(Rewers and Ludvigsson, 2016)	Enterovirus (EV)	Development of β-cell autoimmunity, the presence of enterovirus in pancreatic islets of type 1 diabetic patients.
[84]	(Esposito, S.; 2014)	Herpesviridae, ParvoviridaeTogaviridae, Paramyxoviridae, Retroviridae, Picornaviridae,	Induce islet autoimmunity and ß-cell damage and reduce insulin production, leading to full-blown T1DM.
[86]	(Krogvold et al., 2015)	Enterovirus (EV)	The presence of enterovirus in pancreatic islets of type 1 diabetic patients.
[91]	(Vehik et al., 2019)	Enterovirus A, B, mast adenovirus C, Coxsackievirus, adenovirus	Enterovirus B infections may be involved in the development of islet autoimmunity, but not T1DM; in some young children, coxsackie and adenovirus receptor (*CXADR*) genes independently correlated with islet autoimmunity.
[92]	(Hayakawa et al., 2019)	Coxsackievirus B1	Fulminant T1DM in pregnancy may be associated with Coxsackievirus B1 infection.
[99]	(Butalia et al., 2016)	Mumps, Rubella, Rotavirus, Rnterovirus, Cytomegalovirus	Development of β-cell autoimmunity, molecular mimicry, in vitro, viruses may induce markers of inflammation and alter HLA class I molecule expression.
[101]	(Hodik et al., 2016)	Coxsackievirus	Thecoxsackie–adenovirusreceptor (CAR) is expressed in pancreatic islets of patients with T1DM.
[104]	(Aarnisalo et al., 2008)	Cytomegalovirus (CMV)	Development of beta-cell autoimmunity.
[105]	(Yoneda et al., 2017)	Cytomegalovirus (CMV)	Significantly increased numbers of alpha cells expressing RIG-I and IRF3 development and progression of T1DM.
[106]	(Ekman, et al., 2019)	Cytomegalovirus (CMV)	Development and progression of T1DM.
[107]	(Al-Hakami, 2016)	Cytomegalovirus (CMV)	No correlation between T1DMandvirus infectivity.
[108]	(Nishiumi et al., 2014)	Parvovirus B19	Fulminant type 1 diabetes mellitus associated with parvovirus B 19.
[109]	(Selver Eklioglu et al., 2017)	Parvovirus B19	Diabetic ketoacidosis (DKA) and acute fulminan hepatitis.
[110]	(O’Bryan et al., 2005)	Parvovirus B19	No association between parvovirus B19 infection and the development of T1DM.
[111]	(Honeyman, 2005)	Rotavirus	Molecular mimicry, pancreatic β cell destruction.
[114]	(Harrison et al., 2019)	Rotavirus	Molecular mimicry, development of β-cell autoimmunity.
[115]	(Vaarala et al., 2017)	Rotavirus	Molecular mimicry, development and progression of T1DM.
[116]	(Glanz et al., 2020)	Rotavirus	Molecular mimicry, development of β-cell autoimmunity, rotavirus vaccination does not appear to be associated with T1DM in children.
[118]	(Ramondetti et al., 2012)	Rubella virus, Mumps virus	Mumps and rubella viral infections are associated with T1DM.
[119]	(Korkmaz and Ermiş, 2019)	Rubella virus	Rubella viral infections are associated with T1DM.
[120]	(Gale, 2008)	Rubella virus	Rubella infections predispose to autoimmunity.
[121]	(Vuorinen et al., 1992)	Mumps Virus, Coxsackievirus	In vitro model indicated that mumps and coxsackie B3 viruses infect human fetal pancreatic endocrine cells and are able to alter beta-cell function.
[122]	(Precechtelova et al., 2014)	Human Cytomegalovirus, Parvovirus, Rotavirus, Coxsackievirus, Human Parechovirus, Enteric Cytopathic Human Orphan viruses, Mumps virus, Rubella virus	Persistent infection, molecular mimicry, autoimmune destruction of pancreatic ≤ β-cells, congenital infection, loss of regulatory T-cells. Infection by rubella virus during pregnancy has been related to increased risk of diabetes in the offspring suffering from congenital rubella syndrome.
[123]	(Levet et al., 2017)	Human endogenous retroviruses (HERV)	Pancreatic β cell destruction.
[132]	(Parkkonen et al., 1992)	Mumps virus	The infection is associated with an increase in the expression of HLA class I molecules.
[133]	(Al-Hakami, 2016)	Viracela, measles	Hemagglutinin peptide and Hsp60 peptide induce the cellular immune response; varicella and measles are risk factors in developing type 1 diabetes.
[79]	(Rubino et al., 2020)	SARS-CoV-2	SARS-CoV-2 virus leads to diabetes via binding to its cellular entry—ACE-2 receptors, which are abundant in pancreatic beta cells and adipose tissue, leading to glucose metabolism abnormalities, and pancreatic beta cells destruction.
[75]	(Suwanwongse and Shabarek, 2021)	SARS-CoV-2	The aberrant immunity caused by SARS-CoV-2 may induce an auto-immune attack on the pancreatic islet cells mimicking the pathogenesis of insulin-dependent DM.
[127]	(Lança et al., 2022)	SARS-CoV-2	Delayed diagnosis, low socioeconomic status, and infection have been associated with diabetic ketoacidosis (DKA) in type 1 diabetes mellitus.

**Table 3 biology-11-00608-t003:** The role of bacteria and yeast-like fungi in the development of type 1 diabetes.

No.	Authors	Bacteria and Yeast-Like Fungi	Consequences
Bacteria			
[74]	(Michalska et al., 2019)	*Micrococcus luteus*, *M. lylar*, *Sarcina luteaKocuriarosea*, *Staphylococcus aureus*, *S. epidermidis*, *S. saprophyticus*, *Bacillussubtilis*, *B. cereus*, *B. mycoides*, *B. macerans*, *Pseudomonas aeruginosa*, *Citrobacter freundi*, *Enterobacter aerogenes*, *Aeromonas hydrophila*	A relationship between the number of children with recently diagnosed T1DM and the mean concentration of psychrophilic and mesophilic bacteria in the Pomeranian and Lubelskie Voivodeships (*p* < 0.001).
[125]	(Niegowska et al., 2019)	*Mycobacterium avium* *paratuberculosis* (MAP)	MAP infection leading to HERV-W antigen expression and enhancing the production of autoantibodies in T1D.
**Mold fungi**			
[74]	(Michalska et al., 2019)	*Rhodotorulamucilaginosa*, *Penicillium chrysogenum*, *P. viridicatum*, *Aspergillus niger*, *A. flavus*, *Alternaria alternata*, *Mucor mucedo*, *Rhizopus nigricans*, *Geotrychum* sp., *Biopolaris* sp., *Chrysosporium* sp., *Paecilomyces* sp.	A significant relationship between the number of new cases of T1DM in children and the mean concentration of fungi in bioaerosols in the Lubelskie Voivodeship (*p* < 0.001) but not in the Pomeranian Voivodeship (p = NS).
[140]	(Michalska et al., 2017)	*Penicillium chrysogenum*, *P. viridicatum*, *Aspergillus niger*, *A. flavus*, *Alternaria alternata*, *Mucor mucedo*, *Rhizopus nigricans*, *Geotrychum* sp., *Biopolaris* sp., *Chrysosporium* sp., *Paecilomyces* sp.	A relation between the number of new cases of T1DM and the number of mold fungi (β = 2.923; *p* < 0.001).
[143]	(McCrory et al., 2014)	*Mucor* sp.	Mucormycosis in children with poorly controlled diabetes and ketoacidosis.
[144]	(Dayal et al., 2015)	*Mucor* sp.	Mucormycosis may extend into the orbit and brain and result in high mortality in children with T1DM.
[145]	(Pennell et al., 2018)	*Rhizopus* sp., *Rhizomucor* sp., *Apophysomyces*	Mucormycosis in children with T1DM.

## Data Availability

Not applicable.

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
