# Peer review of "Environmental Factors and the Risk of Developing Type 1 Diabetes—Old Disease and New Data"

_biology, 2022, doi:10.3390/biology11040608_

Round 1

Reviewer 1 Report

Thank you for submitting this work for review. The subject matter is of high intrest and you have included interesting articles.

However:

  1. Extensive editing of English language and style is required, as illustrated by the few examples below:
    1. Line 23-24 The sentence is not a full sentence. Kindly, reformulate/correct
    2. Line 140-141: unclear sentence. Kindly, reformulate
    3. What is meant by “search for materials”? is it scientific articles/ studies?
    4. Line 146-147: “shift in perceptions about the importance of vitamin D.” do you mean “increased interest about the role of vitamin D”?
    5. Line 154: Kindly, correct “new borns” to “newborns”
  2. The Methods of the review is not described. A description of methods is needed, even if the review is not a systematic review. The abstract mentions the databases used but this not enough for the reviewer to judge the quality of the search, the identification of relevant studies, the inclusion/ exclusion criteria of articles, the selection of studies for inclusion and the evaluation of the strengths and limitations of the articles used.
  3. The discussion section is lacking. What are the strength and limitations of your review? What are your recommendations?

Hopefully these comments can help you improve your manuscript.

Author Response

Reviewer 1

Thank you for submitting this work for review. The subject matter is of high intrest and you have included interesting articles.

Dear Reviewer

First of all, the authors would like to thank for your positive opinion as well as questions and suggestions. We tried our best to complete the manuscript in accordance with all comments. A detailed explanation to your queries are presented below. 

Extensive editing of English language and style is required, as illustrated by the few examples below:

Line 23-24 The sentence is not a full sentence. Kindly, reformulate/correct

Author’s Response: Thank you for your remark. In the revised version of the manuscript we deleted that sentence. In addition, the manuscript has been revised, proofread and language-edited to improve its quality

Line 140-141: unclear sentence. Kindly, reformulate

Author’s Response: Thank you for your remark. In the revised version of the manuscript, we changed accordingly:

Interestingly, recent research has shown that excessive weight gain in the first years of life causes growth and the accumulation of both fat and muscle cells [53].

What is meant by “search for materials”? is its scientific articles/ studies?

Author’s Response: In the revised version of the manuscript we changed and it is now: “online search for scientific research”. In addition, detailed methodology in regards to choosing the studies for this review had been now included in the abstract

Line 146-147: “shift in perceptions about the importance of vitamin D.” do you mean “increased interest about the role of vitamin D”?

Author’s Response: Thank you for your remark. In the revised version of the manuscript, we changed it accordingly: “Over the past decade, there has been a significant increase in scientific research focused on the effects of vitamin D on human health”.

Line 154: Kindly, correct “new borns” to “newborns

Author’s Response: Thank you for your remark. In the revised version of the manuscript we changed: It is now: “newborns”

  1. The Methods of the review is not described. A description of methods is needed, even if the review is not a systematic review. The abstract mentions the databases used but this not enough for the reviewer to judge the quality of the search, the identification of relevant studies, the inclusion/ exclusion criteria of articles, the selection of studies for inclusion and the evaluation of the strengths and limitations of the articles used

Author’s Response: Thank you for your remark. In the revised version of our manuscript, we added a description of the methods on page 2, section 1.

Therefore, an extensive online search for scientific research on factors related to diabetes has been performed for the identification of new factors of unexplained etiology. Three databases were searched: PubMed, Cochrane, and Embase. The following ‘methodological filter’ was applied when surveying the literature. Firstly, a comprehensive search of peer-reviewed journals, excluding conference papers, was completed based on a wide range of key terms, including the following keywords: type 1 Diabetes Mellitus, T1DM, infection, viral infection, bacterial infection, fungal infection, microbiome, mold, environment, and environmental. Latin names of each of the pathogens found by the use of the previous keywords were also searched. Secondly, the reference section for each article found was also searched to find additional manuscripts. The search results were limited to the past 11 years of research and discovered 143 manuscripts published between 2011-2022. Additionally, 21 manuscripts from between 2000-2010 and 3 manuscripts from 1974-2000 were referenced for historical reference, as first studies showcasing a certain phenomenon or mechanism. Eligibility criteria chosen for the study were the following: 1) studies published in the English language, between 2000-2022, focused on children, adolescents, or pregnant women (due to the T1DM manifesting itself in the first years of life). Exclusion criteria included: primary focus on type 2 diabetes with only a reference to type 1, short reports/conference articles without methodology or results section, studies published in a language other than English. Animal studies and cell-culture studies were only referenced to explain the mechanism behind a particular relation seen in human studies.

  1. The discussion section is lacking. What are the strength and limitations of your review? What are your recommendations?

Author’s Response: Thank you for your remark. In the revised version of the manuscript, we added a section:

Limitations and Recommendations

The present a systematic review has some limitations that we would like to address. Firstly, our review did not include patients with diabetes type 2 and other types of diabetes. Secondly, the manuscript would be more powerful if a meta-analysis was performed.

Regarding the recommendations, we believe that demonstrating the relation between environmental pollution, infectious diseases (viral, bacterial, fungal) and the presence of type 1 diabetes, makes it necessary to consider broader activities related to the prevention of T1DM. For viral infections such as measles, rubella and mumps, vaccine is the most effective way to fight and control[168][169]. In the context of, reports about new outbreaks these viral diseases, both in children and adults, the situation is becoming disturbing. It is estimated that this increasing of number of cases is influenced by anti-vaccination movements and disinformation in the internet [170]. Therefore, further research and increased preventive measures are necessary. Especially vaccination and educating the public in the context of all possible consequences of developing rubella, measles and mumps. It is also necessary to take a holistic approach to the links between environmental pollution, viral diseases and diabetes. Especially due to the fact that the content of PM2.5 and PM10 in the air is also associated with the occurrence of mumps, as indicated by studies carried out in China [171]. The second important recommendation is to pay attention to diet and the influence of the gut microbiome. Introducing more products containing probiotic bacteria as order to diversify the gut microbiome into the diet seems to be particularly important [154][166][172]  Therefore, due to the complex mechanisms influencing the development of type 1 diabetes it is important to take a holistic approach to this problem from the perspective of environmental pollution, microbiology, immunobiology and dietetics.

However, the section: Discussion has not been added. The primary purpose of the study was to showcase the research dealing with environmental/ viral, bacterial, fungal triggers for T1DM. Since those areas of research are often separate from one another, our review showcases them together in one place. However, we did not compare different results to each other in terms of their specific data etc. In the end, our manuscript is about systematic review and is not a quantitative review that analyzes quantitative results in a quantitative manner. Such works become "meta-analysis" in which we there are distinguish sections: introduction, material and methods, including statistical methods, results and discussion etc. 

  1. Cook DA, West CP (2012) Conducting systematic reviews in medical education: a stepwise approach. Med Educ 46: 943–952

2.Ten Simple Rules for Writing a Literature Review https://journals.plos.org/ploscompbiol/article?id=10.1371/journal.pcbi.1003149

In the journal Biology, there is no discussion in the review manuscripts and the last point concerns “Conclusions” or "Concluding Remarks and Perspectives" or "Practical Implications and Future Directions" etc.

Some examples are given below

https://www.mdpi.com/2079-7737/11/1/22/htm

Clinical Significance of COVID-19 and Diabetes: In the Pandemic Situation of SARS-CoV-2 Variants including Omicron (B.1.1.529)

Conclusions

Rhizosphere Tripartite Interactions and PGPR-Mediated Metabolic Reprogramming towards ISR and Plant Priming: A Metabolomics Review

https://www.mdpi.com/2079-7737/11/3/346/htm

the last point concerns:

Concluding Remarks and Perspectives

https://www.mdpi.com/2079-7737/11/2/339/htm

The Impact of Obesity, Adipose Tissue, and Tumor Microenvironment on Macrophage Polarization and Metastasis

https://www.mdpi.com/2079-7737/11/2/288/htm

Gastrointestinal Incretins—Glucose-Dependent Insulinotropic Polypeptide (GIP) and Glucagon-like Peptide-1 (GLP-1) beyond Pleiotropic Physiological Effects Are Involved in Pathophysiology of Atherosclerosis and Coronary Artery Disease—State of the Art

Practical Implications and Future Directions

or

Concluding Remarks

https://www.mdpi.com/2079-7737/11/2/249/htm

Relevance of miR-223 as Potential Diagnostic and Prognostic Markers in Cancer 

We appreciate your careful evaluation of our work and hope that this revision meets with your approval.

Reviewer 2 Report

Zorena and coworkers in their review article discuss the Environmental factors that may be associated with type 1 diabetes. The review describes the conventional etiology of T1D and then further elaborated on other factors such as viral, bacterial, and fungal infections that maybe linked to T1D. It is an interesting review and is well within the scope of the journal.

Major points:

Microbiota (especially intestinal) is known to influence the outcome of diabetes, especially T1D. Please write a chapter describing the literature that discusses the impact of microbiota on T1D ( in context to environment).

Please cite some mechanistic studies where viral infections have been associated with the onset or acceleration of T1D.

Chapter 2.1: mostly talk about internal factors associated with diabetes. It should be merged with the previous chapter.

Minor:

The manuscript should be vigorously checked for typographical errors.

Line 125: ‘he’ should be ‘the’

Author Response

Reviewer 2

Zorena and coworkers in their review article discuss the Environmental factors that may be associated with type 1 diabetes. The review describes the conventional etiology of T1D and then further elaborated on other factors such as viral, bacterial, and fungal infections that maybe linked to T1D. It is an interesting review and is well within the scope of the journal.

Response to Reviewer 2:

Dear reviewer, thank you for helpful suggestions and comments. The responses to both major and minor points are listed below. Fragments concerning the microbiome and diabetes are added in the manuscript text.

Major points

Reviewer 2:

Microbiota (especially intestinal) is known to influence the outcome of diabetes, especially T1D. Please write a chapter describing the literature that discusses the impact of microbiota on T1D (in context to environment)

Response: Thank you for your comment. The paragraph on linking the composition of the microbiome to the development of diabetes T1D is placed at the end of the section 4. The effect of bacteria and yeast-like fungi on the risk of developing T1DM diabetes.

The correlation between diabetes and bacteria is not only observed in the context of infection. A lot of studies, conducted both on the human subject and mice models indicate a connection between the composition of the intestinal microbiome and diabetes and it focused on both composition and diversity of microorganisms groups [143][144][145]. Research carried out by scientists indicates several potential mechanisms that may explain the impact of changes in the diversity of bacteria in the digestive system and the development of diabetes. However, the exact impact is very complex and has not yet been identified. The most important factors influencing the changes in the microbiome are antibiotic intake, drugs therapy, diet, pH of drinking water, and delivery mode [146] [147]. The composition of the microbiome varies throughout life. It is indicated that in the fall of newborns and infants, delivery mode and method of feeding (breastfeeding or infant formula milk)  have a great impact on intestinal microbiome diversity. Children born by cesarean section have a microbiome composition similar to the mother skin [148]. Research on the development of the human microbiome in infants and the correlation with the development of type 1 diabetes was made by Kostic et al. [149]. They show that in the case of progression of the T1D, the number of microorganisms associated with inflammation increases. They also observed differences in the case of patients with progress and nonprogress of T1D [149]. In the next stages of life, there are more factors influencing the presence of bacteria in the human digestive system. Inadequate diet or drugs can significantly affect microbiome disturbance called dysbiosis, which affects higher insulin resistance and inflammation [150]. Differences in the composition and diversity of microorganisms in the digestive system were observed both in the cases of bacteria and fungi, in studies conducted with patients with T1DM. Research conducted by Kowalewska et al. shows that patients with T1DM show higher species diversity of the yeast-like fungi in stool samples. Moreover, fungal species in patients with T1DM turn out to be more resistant to antifungal treatment [83]. Research on the intestinal microbiome in the context of T1DM focuses on the 3 main issues: reduction of the diversity of microbiota, disturbances in the Firmicutes / Bacteroides ratio, and the reduced number of probiotic bacteria such as lactobacillus [151].  In the studies carried out by Patterson et al., it was also shown that dysbiosis may manifest itself in a reduced differentiation of the microbiome, which in turn increases the predisposition to T1DM development [152]. Moreover, numerous studies show that the composition of the gut microbiome can influence the metabolism of carbohydrates, fatty acids, or amino acids which indirectly may impact the formation of metabolic disorders and diabetes [153][154][155][156]. The development of type 1 diabetes is largely correlated with the diet we eat. A diet high in sugars and fats can accelerate or even induce the changes that cause this disease [157]. Studies conducted by Calcinaro et al. show that oral probiotics supplementing the diet of non-obese mice with probiotics may inhibit the development of auto-immune diabetes [158]. Increasing inflammation is also an important way for developing diabetes. Studies conducted by Pahwa et al., show a significant correlation between the composition of the gut microbiome and inflammation in the presence of diabetes and diabetic complications [159]. Studies on rat models also show a relation between diabetes and the composition of the gut microbiome. In studies published by Roesh et al. it was observed that in the case of bio-breeding diabetes-prone (BB-DP) rats, higher numbers of bacteria of the Bacteroides, Ruminococcus and Eubacterium families were observed in stool samples [160]. A higher number of bacteria that may have a probiotic (Lactobacillus spp. Bifidobacterium spp.) effect was observed in bio-breeding diabetes-resistant (BB-DR) rats [161]. Important mechanism with a potential influence on the development of type 1 diabetes is the molecular mimicry of proteins produced by intestinal bacteria. Some of the proteins produced by the intestinal microbiome have a molecular structure similar to self-anti-gene in pancreas. An example of such protein is the Mpt protein produced by the Leptotrichia goodfellowii bacteria of the Fusobacteria family [162]

Reviewer 2:

Please cite some mechanistic studies where viral infections have been associated with the onset or acceleration of T1D

Author’s Response: Thanks for your question. Below are some examples:

In studies conducted by Krogvold et al., enteroviruses were detected in the pancreas of patients with newly di-agnosed type 1 diabetes [86].

[86]       Krogvold, L.;Edwin, B.;Buanes, T.;Frisk, G.;Skog, O.;Anagandula, M.;Korsgren, O.;Undlien, D.;Eike, M.C.;Richardson, S.J.;Leete, P.;Morgan, N.G.;Oikarinen, S.;Oikarinen, M.;Laiho, J.E.;Hyöty, H.;Ludvigsson, J.;Hanssen, K.F. and Dahl-Jørgensen, K., Detection of a Low-Grade Enteroviral Infection in the Islets of Langerhans of Living Patients Newly Diagnosed With Type 1 Diabetes, Diabetes, vol. 64, no. 5, pp. 1682 LP – 1687, May 2015, doi: 10.2337/db14-1370.

Also, the research by Hodik et al., confirmed the increased level of CAR receptor (Coxsackie and adenovirus receptor) expression on pancreatic cells in patients diagnosed with type 1 diabetes [101].

[101]  Hodik, M.;Anagandula, M.;Fuxe, J.;Krogvold, L.;Dahl-Jørgensen, K.;Hyöty, H.;Sarmiento, L.;Frisk, G. and Consortium, =POD-V, Coxsackie-adenovirus receptor expression is enhanced in pancreas from patients with type 1 diabetes, BMJ open diabetes research & care, vol. 4, no. 1, p. e000219, 2016. doi: 10.1136/bmjdrc-2016-000219.

In the research work of Honkanen et al., the material for the isolation of enterovirus strains was faeces collected from children aged 3 months to 2-3 years. The studies confirmed the observations made by other scientists that enteroviral infections may play a role in initiating the process of autoimmune destruction of beta cells. Researchers also found that viral infections occurred even a year before insulin production disorders were diagnosed [90].

[90]    Honkanen, H.;Oikarinen, S.;Nurminen, N.;Laitinen, O.H.;Huhtala, H.;Lehtonen, J.;Ruokoranta, T.;Hankaniemi, M.M.;Lecouturier, V.;Almond, J.W.;Tauriainen, S.;Simell, O.;Ilonen, J.;Veijola, R.;Viskari, H.;Knip, M. and Hyöty, H., Detection of enteroviruses in stools precedes islet autoimmunity by several months: possible evidence for slowly operating mechanisms in virus-induced autoimmunity, Diabetologia, vol. 60, no. 3, pp. 424–431, 2017, doi: 10.1007/s00125-016-4177-z.

Recently, a clinical case of a 31-year-old pregnant patient suspected of having type 1 diabetes was described (Fulminant type 1 diabetes mellitus). Tests for the presence of antibodies in the patient's serum confirmed infection with Coxsackievirus B1 [92].

[92]    Hayakawa, T.;Nakano, Y.;Hayakawa, K.;Yoshimatu, H.;Hori, Y.;Yamanishi, K.;Yamanishi, H.;Ota, T. and Fujimoto, T., Fulminant type 1 diabetes mellitus associated with Coxsackievirus type B1 infection  during pregnancy: a case report., J. Med. Case Rep., vol. 13, no. 1, p. 186, Jun. 2019, doi: 10.1186/s13256-019-2130-8.

American scientists, on the other hand, conducted in vitro studies confirming whether human β stem cells (SC-β) could serve as a model for studying the effect of viral infection on TDM1 [102]. The human β stem cells (SC-β) used for the study were derived from the embryonic stem cell line HUES8 and were infected with Coxackie virus strain JVB CVB4. Studies have shown that SC-β cells can serve as a model for T1DM development. The same authors proved, that CVB infection strongly induces innate immune responses, transcriptional decreases in insulin, and impaired insulin secretion [102].

[102]  Nyalwidhe, J.O.;Jurczyk, A.;Satish, B.;Redick, S.;Qaisar, N.;Trombly, M.I.;Vangala, P.;Racicot, R.;Bortell, R.;Harlan, D.M.;Greiner, D.L.;Brehm, M.A.;Nadler, J.L. and Wang, J.P., Proteomic and Transcriptional Profiles of Human Stem Cell-Derived β Cells Following  Enteroviral Challenge., Microorganisms, vol. 8, no. 2, Feb. 2020, doi: 10.3390/microorganisms8020295.

In vitro studies, on the other hand, suggest that duct cells may be the site where enterovirus survives, allowing it from time to time to spread to pancreatic beta cells [103].

[103]  Lietzén, N.;Hirvonen, K.;Honkimaa, A.;Buchacher, T.;Laiho, J.E.;Oikarinen, S.;Mazur, M.A.;Flodström-Tullberg, M.;Dufour, E.;Sioofy-Khojine, A.-B.;Hyöty, H. and Lahesmaa, R., Coxsackievirus B Persistence Modifies the Proteome and the Secretome of Pancreatic  Ductal Cells., iScience, vol. 19, pp. 340–357, Sep. 2019, doi: 10.1016/j.isci.2019.07.040.

In other studies, the authors showed a strong correlation between the cytomegalovirus (CMV) genome detectable in lymphocytes and autoantibodies against pancreatic islets in patients with newly diagnosed type 1 diabetes [104].

[104]  Aarnisalo, J.;Veijola, R.;Vainionpää, R.;Simell, O.;Knip, M. and Ilonen, J., Cytomegalovirus infection in early infancy: risk of induction and progression of  autoimmunity associated with type 1 diabetes., Diabetologia, vol. 51, no. 5, pp. 769–772, May 2008, doi: 10.1007/s00125-008-0945-8.

Also, Yoned et al., who carried out histopathological examination of the pancreas within a study that showed damage to β-cells as a result of CMV infection [105].

[105]  Yoneda, S.;Imagawa, A.;Fukui, K.;Uno, S.;Kozawa, J.;Sakai, M.;Yumioka, T.;Iwahashi, H. and Shimomura, I., A Histological Study of Fulminant Type 1 Diabetes Mellitus Related to Human Cytomegalovirus Reactivation, J. Clin. Endocrinol. Metab., vol. 102, no. 7, pp. 2394–2400, Jul. 2017, doi: 10.1210/jc.2016-4029.

Another study by Saber and Mohammed also indicated a significant role of cytomegalovirus and Epstein-Barr virus in the pathogenesis of type 1 diabetes [47]. The researchers studied 56 patients aged 3 to 22. Out of those, 53 (94.60%) patients with T1DM were positive for IgG antibodies against cytomegalovirus, and 24 (42.90%) for IgG antibodies against Epstein Barr virus compared to the control group [47].

[47]    Ibrahim, Saber A-Z A-B.; Mohammed, A.H. (2019): The role of Human Cytomegalovirus and Epstein-Barr virus in type 1 Diabetes Mellitus, Ann Trop & Public Health; 22(9): S267. DOI: http://doi.org/10.36295/ASRO.2019.220912

Eklioğlu et al. described the case of a 9-year-old boy who was diagnosed with type 1 diabetes 3 years earlier, and who was hospitalized due to ketoacidosis (DKA) and acute hepatitis caused by B19 parvovirus [109].

[109]  Selver Eklioglu, B.;Atabek, M.;Akyürek, N. and Gümüş, M., Parvovirus Infection in a Child Complicated with Diabetic Ketoacidosis and Acute Fulminant Hepatitis: a Case Report, J. Pediatr. Infect., vol. 11, pp. 92–94, Jun. 2017, doi: 10.5578/ced.57493.

The fact is that in both the SARS-CoV epidemic and the SARS-CoV-2 pandemic, the presence of the virus was demonstrated in the pancreas of deceased patients [130][131]. Research conducted by a Chinese team of researchers during the SARS-CoV coronavirus epidemic showed a significant link between pancreatic cells and the presence of the virus's genetic material. Analyzes were performed on postmortem material using RT-PCR and focused on the detection of the M protein genes [131]. Similarly, research was also carried out during the current pandemic by scientists from Germany [130]. Based on post-mortem studies of 11 patients who died from COVID-19, the presence of SARS-CoV2 genetic material in the pancreas of COVID-19 patients was also confirmed [130].

[130]  Deinhardt-Emmer, S.;Wittschieber, D.;Sanft, J.;Kleemann, S.;Elschner, S.;Haupt, K.F.;Vau, V.;Häring, C.;Rödel, J.;Henke, A.;Ehrhardt, C.;Bauer, M.;Philipp, M.;Gaßler, N.;Nietzsche, S.;Löffler, B. and Mall, G., Early postmortem mapping of SARS-CoV-2 RNA in patients with COVID-19 and the correlation with tissue damage, Elife, vol. 10, Mar. 2021, doi: 10.7554/eLife.60361.

[131]  Ding, Y.;He, L.;Zhang, Q.;Huang, Z.;Che, X.;Hou, J.;Wang, H.;Shen, H.;Qiu, L.;Li, Z.;Geng, J.;Cai, J.;Han, H.;Li, X.;Kang, W.;Weng, D.;Liang, P. and Jiang, S., Organ distribution of severe acute respiratory syndrome(SARS) associated coronavirus(SARS-CoV) in SARS patients: implications for pathogenesis and virus transmission. J Pathol. 2004 Jun;203(2):622-30. doi: 10.1002/path.1560.

Reviewer 2:

Chapter 2.1: mostly talk about internal factors associated with diabetes. It should be merged with the previous chapter.

Author’s Response: Thank you for your suggestions. Chapter 2.1 “The Accelerator Hypothesis, the β-cell “overload” the risk of developing diabetes”  has been moved to the “Introduction”.

1.3. The Accelerator Hypothesis, the β-cell “overload” and the risk of developing diabetes

Minor points

Reviewer 2:

The manuscript should be vigorously checked for typographical errors.

Author’s Response: Thank you for your comment.  The article has been checked for typographical errors.

Reviewer 2:

Line 125: ‘he’ should be ‘the’

Author’s Response: Thank you for your comment.  Page 8: ‘he’ as been changed to the ‘the’.

1.3. The Accelerator Hypothesis, the β-cell “overload” and the risk of developing diabetes

We appreciate your careful evaluation of our work and hope that this revision meets with your approval.

Reviewer 3 Report

This is well reviewed manuscript. The reviewer suggest minor points of this manuscript.

  1. Page 6, L293-> "The researchers studied 56 patients aged 3 to 22 years." Is this sentence explaning the study of Saber and Mohammed? Please make the pharagraph clear.
  2.  Page7, L333-> "Environmemtal fators that may play a role in the pathogenesis of T1DM also include rubella......" please make this sentence clear. Why  suddenly environmental factors?
  3. This review is more focused on virus, bacteria and fungus. And the title of this manuscript is "environmental factors and the risk of developing T1DM". Please add the list of the reviewed article of environmental factors as a table.
  4. In page 13, L496 " A better understanding of the role of environmetal factors to the risk of TIDM in children,,,,,,,is of great importance for people and families"-. please revise this sentence. This sentence is far from the content of this manuscript and is not appropriate for the conclusion. 

Author Response

Reviewer 3

Author’s Response to reviewer 3 comments:

Dear reviewer, we are grateful for your valuable and accurate comments. We revised our manuscript in accordance to all your suggestions which greatly improved our manuscript. The answersto the specific comments are presented below and in the tracked version of the manuscript.

Reviewer 3 wrote:

This is well reviewed manuscript. The reviewer suggests minor points of this manuscript.

  1. Page 6, L293-> "The researchers studied 56 patients aged 3 to 22 years." Is this sentence explaining the study of Saber and Mohammed? Please make the paragraph clear.

Author’s Response: Thank you for this comment. Indeed, the sentence was not clear due to the unfortunate spacing between lines. We rephrased it, you can see the new version down below:

Page: 8

Another study by Saber and Mohammed also indicated a significant role of cytomegalovirus and Epstein-Barr virus in the pathogenesis of type 1 diabetes. The researchers studied 56 patients aged 3 to 22. Out of those, 53 (94.60%) patients with T1DM were positive for IgG antibodies against cytomegalovirus, and 24 (42.90%) for IgG antibodies against Epstein Barr virus compared to the control group.

Reviewer 3 wrote:

  1.  Page7, L333-> "Environmental factors that may play a role in the pathogenesis of T1DM also include rubella......" please make this sentence clear. Why suddenly environmental factors?

Author’s Response: Thank you for that comment. The sentence was rephrased:

Page: 9

Other viral factors that may play a role in the pathogenesis of T1DM also include rubella, mumps, and measles viruses [118][119][120].

Reviewer 3 wrote:

  1. This review is more focused on virus, bacteria and fungus. And the title of this manuscript is "environmental factors and the risk of developing T1DM". Please add the list of the reviewed article of environmental factors as a table.

Author’s Response: Thank you for that comment. We added Table 1.

Table 1. The climatic conditions, vitamin D deficiency, cow's milk feeding and the risk of developing type 1 diabetes

No.

Authors

Climatic conditions, lack of vitamin D deficiency, cow's milk feeding

Consequences

56

Waernbaum and Dahlquist 2016

climate/ air temperature and the number of sunshine hours

low mean temperature rather than few sunshine hours are associated with an increased incidence of type 1 diabetes in children

63

Sørensen et al., 2016

vitamin D deficiency in pregnancy

low level of 25 (OH) D during the third trimester of pregnancy may associated with a higher risk of developing diabetes in the offspring

64

Jacobsen et al., 2016;

sun exposure/ Vitamin D deficiency in pregnancy

higher sun exposure in the third trimester of pregnancy is associated with a lower risk (HR) of T1D at 5 to 9 years of age

67

Mohr et al.

intensity of UVB radiation and the number of hours of sunshine per day

the low intensity of UVB radiation at higher latitudes in both hemispheres (R2 for latitude = 0.25, p <0.0001) and the number of hours of sunshine per day were associated with a much higher incidence of T1D in childhood

69

Abela, and Fava, 2019

climatic conditions

a significant negative association was found between T1D incidence and shortest distance from sea, mean hours of sunshine, and mean temperature, and a positive association was found between T1D incidence and latitude.

70

Ahadi et al, 2011

association between environmental factors and risk of type 1 diabetes

maternal age > 35 years at delivery, duration of > 6 months of cow milk feeding, lack of vitamin D supplementation and caesarean delivery were significantly associated with diabetes type 1

88

Kostopoulou, et.al., 2021

seasonal variation of T1DM onset

a seasonality of the T1DM diagnosis is shown, with a predominance of the colder months of the year.

Reviewer 3 wrote:

  1. In page 13, L496 " A better understanding of the role of environmental factors to the risk of TIDM in children,,,,,,,is of great importance for people and families"-. please revise this sentence. This sentence is far from the content of this manuscript and is not appropriate for the conclusion. 

Author’s Response: Thank you for this comment. Indeed, the phrase was revised.

Page: 17

A better understanding of the role of viral, bacterial and yeast-like fungi infections related to the risk of T1DM in children and adolescents and the identification of new risk factors, especially those spread by the droplet route, is of great importance for people and families with diabetes.

We appreciate your careful evaluation of our work and hope that this revision meets with your approval.

Round 2

Reviewer 1 Report

Thank you for responding to my previous comments and the effort put into this work.

I remain not convinced with the methodology. From my point of view, it is a literature review and do not fulfill 

the criteria for a systematic review. I encourage you to visit Cochrane.org and https://jbi.global for details on conducting a systematic review. A meta-analysis can be part of a systematic review, but it is not a must. However, a review whether systematic, review of reviews (umbrella review), a scoping review each has specific methodology that you need to overall follow for the clarity of the results.

Author Response

Response to Reviewer 1 comment:

Reviewer wrote:

Thank you for responding to my previous comments and the effort put into this work.

I remain not convinced with the methodology. From my point of view, it is a literature review and do not fulfill the criteria for a systematic review. I encourage you to visit Cochrane.org and https://jbi.global for details on conducting a systematic review. A meta-analysis can be part of a systematic review, but it is not a must. However, a review whether systematic, review of reviews (umbrella review), a scoping review each has specific methodology that you need to overall follow for the clarity of the results.

Response:

Thank you for your comment. We visited Cochrane.org and https://jbi.global websites. According to jbi.global, we followed a protocol tailored and designed for "Systematic Prevalence and Incidence Reviews". It is worth noting that the number of possible approaches and guidelines regarding the preparation of review manuscript is growing and there is no one “fit-to-all” approach. However, we do agree that the following steps are required for a systematic review of any type:

  1. Formulating a review question/ review objective
  2. Defining inclusion and exclusion criteria
  3. Preparation of search strategy
  4. Data extraction
  5. Analyzing and synthesizing the relevant studies
  6. Presenting and interpreting results

In accordance to those steps, we formulated our methodology steps in the introduction part:

  1. Review question/review objectives:

Despite many studies, the risk factors of type 1 diabetes in children and adolescents are not fully understood and remain a big challenge for researchers -> Therefore, an extensive research search of factors related to diabetes has been performed for the identification of factors of unexplained etiology.

  1. Defining inclusion and exclusion criteria

Eligibility criteria chosen for the study were the following: 1) studies published in the English language, between 2000-2022, focused on children, adolescents, or pregnant women (due to the T1DM manifesting itself in the first years of life). Exclusion criteria included: primary focus on type 2 diabetes with only a reference to type 1, short reports/conference articles without methodology or results section, studies published in a language other than English. Animal studies and cell-culture studies were only referenced to explain the mechanism behind a particular relation seen in human studies.

  1. Preparation of search strategy/data extraction

Three databases were searched: PubMed, Cochrane, and Embase. The following ‘methodological filter’ was applied when surveying the literature. Firstly, a comprehensive search of peer-reviewed journals, excluding conference papers, was completed based on a wide range of key terms, including the following keywords: type 1 Diabetes Mellitus, T1DM, infection, viral infection, bacterial infection, fungal infection, microbiome, mold, environment, and environmental. Latin names of each of the pathogens found by the use of the previous keywords were also searched. Secondly, the reference section for each article found was also searched to find additional manuscripts. The search results were limited to the past 11 years of research and discovered 143 manuscripts published between 2011-2022. Additionally, 21 manuscripts from between 2000-2010 and 3 manuscripts from 1974-2000 were referenced for historical reference, as first studies showcasing a certain phenomenon or mechanism.

  1. Analyzing and synthesizing the relevant studies/ Presenting and interpreting results

The whole-body of the manuscript.

Therefore, to our knowledge, we have used methodological tools appropriate for writing review articles. What is more, much of the data in this manuscript relates to our own research - I have been researching diabetes for over 30 years. I would also like to add that, I am an academic teacher and teach the Methodology of Scientific Research module. I am familiar with many diverse methodological tools. From my point of view,  the number of possible approaches and guidelines regarding the preparation of review manuscript is growing and there is no one fit-to-all approach.

We hope that you will accept our methods and manuscript.

Kind regards

Reviewer 2 Report

The authors have incorporated all the suggested changes meticulously. The manuscript is much improved upon revision and may be accepted for publication after correcting minor typographical errors. I congratulate the team for the excellent work.

Author Response

Response to Reviewer 2:

Reviewer wrote:

The authors have incorporated all the suggested changes meticulously. The manuscript is much improved upon revision and may be accepted for publication after correcting minor typographical errors. I congratulate the team for the excellent work.

Response:

Thank you for your kind review. We proofread the manuscript to remove any doble-spaces and punctuation mistakes. Thank you.

Kind regards